# Word embeddings for application in geosciences: development, evaluation and examples of soil-related concepts

José Padarian and Ignacio Fuentes

Sydney Institute of Agriculture & School of Life and Environmental Sciences, The University of Sydney, New South Wales, Australia

**Correspondence:** José Padarian (jose.padarian@syndey.edu.au)

**Abstract.** A large amount of descriptive information is available in geosciences. This information is usually considered subjective and ill-favoured compared with its numerical counterpart. Considering the advances in natural language processing and machine learning, it is possible to utilise descriptive information and encode it as dense vectors. These word embeddings, which encode information about a word and its linguistic relationships with other words, lay on a multi-dimensional space where angles and distances have a linguistic interpretation. We used 280,764 full-text scientific articles related to geosciences to train a domain-specific language model capable of generating such embeddings. To evaluate the quality of the numerical representations, we performed three intrinsic evaluations, namely: the capacity to generate analogies, term relatedness compared with the opinion of a human subject, and categorisation of different groups of words. Since this is the first attempt to evaluate word embedding for tasks in the geosciences domain, we created a test suite specific for geosciences. We compared our results with general domain embeddings commonly used in other disciplines. As expected, our domain-specific embeddings (GeoVec) outperformed general domain embeddings in all tasks, with an overall performance improvement of 107.9%. We also presented an example were we successfully emulated part of a taxonomic analysis of soil profiles which was originally applied to soil numerical data, which would not be possible without the use of embeddings. The resulting embedding and test suite will be made available for other researchers to use and expand.

## 1 Introduction

Machine learning (ML) methods have been used in many fields of geosciences (Lary et al., 2016) to perform tasks such as classification of satellite imagery (Maxwell et al., 2018), soil mapping (McBratney et al., 2003), mineral prospecting (Caté et al., 2017), flood prediction (Mosavi et al., 2018). Thanks to their capability to deal with complex nonlinearities present in the data, ML usually outperforms more traditional methods in terms of predictive power. The application of ML in geosciences usually prioritise numerical or categorical data over qualitative descriptions, which are usually considered of subjective nature (McBratney and Odeh, 1997). However, it must be taken into account the resources that have been invested in collecting large

amounts of descriptive information from pedological, geological and other fields of geosciences. Neglecting descriptive data due to its inconsistency seems wasteful, yet natural language processing (NLP) techniques, which involve the manipulation and analysis of language (Jain et al., 2018), have rarely been applied in geosciences.

For soil sciences, NLP opens the possibility to use a broad range of new analyses. Some examples include general, discipline-wide methods such as automated content analysis (Nunez-Mir et al., 2016) or recommendation systems (Wang and Blei, 2011) which can take advantage of the current literature. More specific cases could take advantage of big archives of descriptive data, like the ones reported by Arrouays et al. (2017). The authors mention examples such as the Netherlands with more than 327,000 auger descriptions covering agricultural, forest and natural lands, or the north-central US with 47,364 pedon descriptions covering 8 states.

Approaches to deal with descriptive data include the work of Fonseca et al. (2002) who proposed the use of ontologies to integrate geographic information of different kinds. At the University of Colorado, Chris Jenkins created a structured vocabulary for geomaterials (http://instaar.colorado.edu/~jenkinsc/dbseabed/resources/geomaterials/) using lexical extraction (Miller, 1995), names decomposition (Peckham, 2014) and distributional semantics (Baroni et al., 2012) in order to characterise word terms for use in NLP and other applications. A different approach, perhaps closer to the preferred quantitative methods, is the use of dense word embeddings (vectors) which encode information about a word and its linguistic relationships with other words, positioning it on a multi-dimensional space. The latter is the focus of this study.

There are many general-purpose word embeddings trained on large corpora from social media or knowledge organisation archives such as Wikipedia (Pennington et al., 2014; Bojanowski et al., 2016). These embeddings have been proven to be useful in many tasks such as machine translation (Mikolov et al., 2013a), video description (Venugopalan et al., 2016), document summarisation (Goldstein et al., 2000), and spell checking (Pande, 2017). However, for field-specific tasks, many researchers agree that word embeddings trained on specialised corpora can capture the semantics of terms better than those trained on general corpora (Jiang et al., 2015; Pakhomov et al., 2016; Roy et al., 2017; Nooralahzadeh et al., 2018; Wang et al., 2018).

As far as we are aware, this is the first attempt to develop and evaluate word embedding for the geosciences domain. This paper is structured as follow: first, we define what word embeddings are, explaining how they work and showing examples to help the reader understand some of their properties. Second, we describe the text data used and the pre-processes required to train a language model and generate these word embedding (GeoVec). Third, we illustrate how a natural language model can be quantitatively evaluated and we present the first test dataset for the evaluation of word embeddings specifically developed for the geosciences domain. Fourth, we present results of an intrinsic evaluation of our language model using our test dataset and we explore some of the characteristics of the multi-dimensional space and the linguistic relationships captured by the model through examples of soil-related concepts. Finally, we present a simple, illustrative example of how the embedding can be used in a downstream task.

## 2 Word embeddings

Word embeddings have been commonly used in many scientific disciplines, thanks to their application in statistics. For example, one-hot encodings (Fig. 1), also know as "dummy variables", have been used in regression analysis since at least 1957 (Suits, 1957). In one-hot encoding, each word is represented by a vector of length equal to the number of classes or words, where each dimension represents a feature. The problem with this representation is that the resulting array is sparse (mostly zeros) and very large when using large corpora, and also presents the problem of poor estimation of the parameters of the less-common words (Turian et al., 2010). A solution for these problems is the use of unsupervised learning to induce dense, low-dimensional embeddings (Bengio, 2008). The resulting embeddings lay on a multi-dimensional space where angles and distances have a linguistic interpretation.

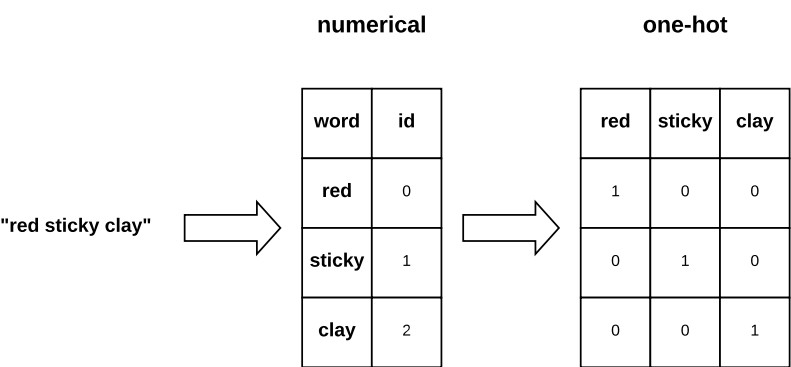

**Figure 1.** Example of two encodings of the phrase "red sticky clay", numerical and one-hot.

These dense, real vectors allow models, specially neural networks, to generalise to new combinations of features beyond those seen during training thanks to the properties of the vector space where semantically related words are usually close to each other (LeCun et al., 2015). Since the generated vector space also has properties such as addition and subtraction, Mikolov et al. (2013b) gives some examples of calculations that can be performed using word embedding. For instance the operation vec("Berlin") - vec("Germany") + vec("France") generates a new vector. When they calculated the distance from that resulting vector to all the words from the model vocabulary, the closest one was the word "Paris". Fig. 2 presents a principal component analysis (PCA) projection of pairs of words with the country-capital relationship. Without explicitly enforcing this relationship when creating the language model, the resulting word embeddings encode the country-capital relationship due to the high co-occurrence of the terms. In Fig. 2 it is also possible to observe a second relationship, geographic location, where South American countries are positioned to the right, European countries in the middle and (Eur-)Asian countries to the left.

Potentially, each dimension and interaction within the generated vector space encodes a different type of relationship extracted from the data. Thanks to the properties of the generated vector space, we give ML algorithms the capacity to utilise and understand text and we are able to use the same methods usually designed for numerical data (e.g. clustering, principal component). In the next sections, we describe how we generated a language model that yields word embeddings that encode

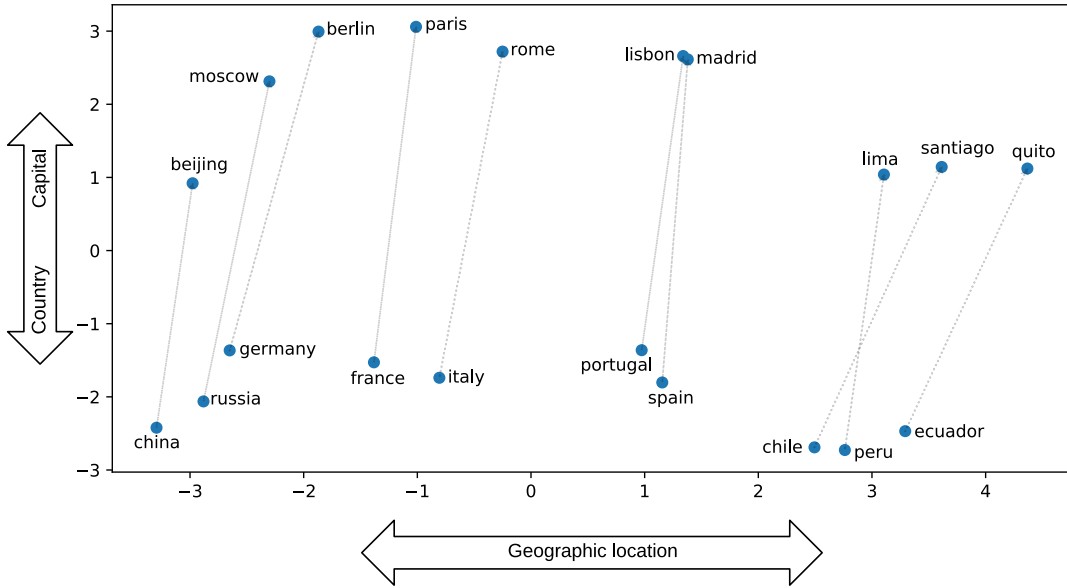

**Figure 2.** Examples of two-dimensional PCA projection of selected word embeddings using a general domain model. The figure illustrates the country-capital relationship learned by the model. Also notice that the model learned about the geographic relationship between the places. Example adapted from Mikolov et al. (2013b).

semantic and syntactic relations specific for the geosciences domain, we visualise some of those relations and we illustrate how to evaluate them numerically.

## 3 Data, text pre-processing and model training

### 3.1 Corpus

5   The corpus was generated by retrieving and processing 280,764 full-text articles related to geosciences. We used the Elsevier ScienceDirect APIs to search for manuscripts that matched the terms listed in Table 1, which cover a broad range of topics. These terms were selected based on their general relationship with geosciences and specifically soil science. We also included Wikipedia articles which list and concisely define some concepts like types of rocks, minerals, and soils, providing more context than a scientific publication, considering that the model depends on words co-occurrences. We down-
10   loaded the text from Wikipedia articles "List_of_rock_types", "List_of_minerals", "List_of_landforms", "Rock_(geology)", "USDA_soil_taxonomy" and "FAO_soil_classification", and also all the Wikipedia articles linked from those pages.

**Table 1.** Search terms used to retrieve full-text articles from Elsevier ScienceDirect APIs.

| Search terms | | |
| --- | --- | --- |
| Acrisol | Geosciences | Permafrost |
| Alfisol | Groundwater | Petrology |
| Allophane | Gypsisols | Podzols |
| Andisol | Histosol | Sedimentary |
| Andosols | Hydrogeology | Sedimentary mineralogy |
| Aridisol | Igneous petrology | Sedimentary petrology |
| Chernozems | Imogolite | Sedimentary rocks |
| Entisol | Inceptisol | Sedimentology |
| Environmental geology | Lithology | Soil classification |
| Field geology | Metamorphic petrology | Spodosol |
| Gelisol | Mineralogy | Stratigraphy |
| Geochemistry | Mollisol | Ultisol |
| Geology | Oxisol | Vertisol |
| Geomaterials | Peatland | Volcanic soils |
| Geomorphology | Pedogenesis | |
| Geophysics | Pedology | |

## 3.2 Pre-processing

The corpus was split into sentences which were then pre-processed using a sequence of commonly used procedures including: *a)* removing punctuation, *b)* lower-casing, *c)* removing digits and symbols, and *d)* removing (easily identifiable) references. The cleaned sentences were then tokenised (split into words). In order to decrease the complexity of the vocabulary, we lemmatised all nouns to their singular form and removed all the words with less than 3 characters. We also removed common English words such as 'the', 'an' and 'most' since they are not discriminating and unnecessarily increase the model size and processing time (a full list of the removed "stop words" can be found in the documentation of the nltk python library (Bird and Loper, 2004)). Finally, we excluded sentences with less than 3 words. The final corpus has a vocabulary size of 701,415 (unique) words and 305,290,867 tokens.

## 3.3 Model training

For this work, we used the GloVe (Global Vectors) model (Pennington et al., 2014), developed by Stanford University NLP group, which achieved great accuracy on word analogy tasks and outperformed other word embedding models on similarity and entity recognition tasks. As many NLP methods, GloVe relays on ratios of word-word co-occurrence probabilities in the corpus. To calculate the co-occurrence probabilities, GloVe uses a local context window, where a pair of words $d$ words apart

contributes to a $1/d$ to the total count. After the co-occurrence matrix $X$ is calculated, GloVe minimises the least-squares problem

$$\sum_{i,j=1}^{V} f(X_{ij}) \left( w_i^T \hat{w}_j + b_i + \hat{b}_j - log X_{ij} \right)^2 \tag{1}$$

where $X_{ij}$ is the co-occurrence between the target words $i$ and the context word $j$, $V$ is the vocabulary size, $w_i$ is the word embedding, $\hat{w}_j$ is a context word embedding, $b_i$ and $\hat{b}_j$ are biases for $w_i$ and $\hat{w}_j$, respectively, and $f(X_{ij})$ is the weighting function

$$f(X_{ij}) = \begin{cases} (X_{ij}/x_{\max})^\alpha & \text{if } X_{ij} < x_{\max} \\ 1 & \text{otherwise} \end{cases} \tag{2}$$

that assures that rare and frequent co-occurrences are not overweighted. Pennington et al. (2014) recommend using the values 0.75 for the smoothing parameter $\alpha$ and 100 for the maximum cutoff count $x_{\max}$.

We trained the model during 60 epochs, where 1 epoch corresponds to a complete pass through the training dataset. During the training phase, we experimented using embedding with different number of components (dimensions) and different context window sizes. Here we present the results for 300 components and a context window of size 10, which represents a good balance between model size, training time and performance.

## 4 Evaluation of word embeddings

Given the characteristic of the vector space, the most common method to evaluate word embeddings is to asses their performance in tasks that test if semantic and syntactic rules are properly encoded. Many studies have presented datasets to perform this task. Rubenstein and Goodenough (1965) presented a set of 65 noun synonyms to test the relationship between the semantic similarity existing between a pair of words and the degree to which their contexts are similar. More recent and larger test datasets and task types have been proposed (Finkelstein et al., 2002; Mikolov et al., 2013c; Baroni et al., 2014) but they all have been designed to test general domain vectors. Because this work aims to generate embeddings for the geosciences domain, we developed a test suite to evaluate their intrinsic quality in different tasks, which are described below.

**Analogy:** Given two related pairs of words, $a$:$b$ and $x$:$y$, the aim of the task is to answer the question "$a$ is to $x$ as $b$ is to?". The set includes 50 quartets of words with different levels of complexity, from simple semantic relationships to more advance syntactic relations. In practice, it is possible to find $y$ by calculating the cosine similarity between the differences of the paired vectors:

$$\frac{(v_b - v_a) \cdot (v_y - v_x)}{\|v_b - v_a\| \|v_y - v_x\|} \tag{3}$$

In this case, $v_y$ is the embedding for each word of the vocabulary and $y$ is the word with the highest cosine similarity. Some examples of analogies are: "moraine is to glacial as terrace is to ____? (fluvial)", "limestone is to sedimentary as tuff is to ____? (volcanic)" and "chalcantite is to blue as malachite is to ____? (green)".

We estimated the top-1, top-3, top-5 and top-10 accuracy score, recording a positive result if $y$ was within the first 1, 3, 5 or 10 words returned by the model, respectively.

**Relatedness:** For a given pair of words $(a, b)$, a score of 0 or 1 is assigned by a human subject if the words are unrelated or related, respectively. The set includes 100 pairs of scored pairs of words. The scores are expected to have a high correlation with the cosine similarity between the embeddings of each pair of words. In this work, we used the Pearson correlation coefficient to evaluate the model against annotations made by 3 people with a geosciences background.

**Categorisation:** Given 2 sets of words $s_1 = \{a, b, c, ...\}$ and $s_2 = \{x, y, z, ...\}$, this test should be able to correctly assign each word to its corresponding group using a clustering algorithm. We provide 30 tests with 2 clusters each. We estimated the v-measure score (Rosenberg and Hirschberg, 2007), which takes into account the homogeneity and completeness of the clusters, after projecting the multi-dimensional vector space to a two-dimensional PCA space and performing a k-means clustering. Given that k-means is not deterministic (when using random centroids initiation), we used the mean v-measure score of 50 realisations.

We compared our results with general domain vectors trained on Wikipedia articles (until 2014) and the Gigaword v5 catalogue, which comprise 6 billion tokens and is provided by the authors of GloVe at https://nlp.stanford.edu/projects/glove/.

## 5 Illustrative example

In order to illustrate the use of word embedding in a downstream application, we decided to emulate part of the analysis of a soil taxonomic system performed by Hughes et al. (2017). They used 23 soil variables (e.g. sand content and bulk density), in their majority numerical and continuous except for two binary variables representing the presence or absence of water or ice. Those variables correspond to the representation of horizons from soil profiles, which were then aggregated (mean) at different taxonomic levels to obtain class centroids.

Our analysis was similar, but, instead of using soil variables, we used the word embedding corresponding to the textual description of 10,000 soil profile descriptions downloaded from the USDA-NRCS Web Site for Official Soil Series Descriptions and Series Classification. The descriptions were pre-processed using the same pipeline used for the corpus (Section 3.2). After obtaining the embeddings for each token in the descriptions, we calculated the mean values per profile, which can be considered as an embedding at the profile level. The profiles and their corresponding 300-dimensional embeddings were aggregated at Great Group (GG) level (Soil Taxonomy) and a mean embedding value was estimated (equivalent to the centroids obtained by Hughes et al. (2017)). After projecting the GG embeddings into a 2-dimensional PCA space, we computed the convex-hull per soil order (smaller convex polygon needed to contain all the GG points for a particular soil order) as a way of visualising their extent.

# 6 Results and discussion

## 6.1 Co-occurrence

Before training the language model, the first output of the process is a co-occurrence matrix. This matrix encodes useful information about the underlying corpus (Heimerl and Gleicher, 2018). Fig. 3 shows the co-occurrence probabilities of soil taxonomic orders and some selected words. It is possible to observe that concepts generally associated with a specific order co-occur in the corpus, such as soil cracks, which are features usually present in Vertisols; or Andisols being closely related to areas with volcanic activity.

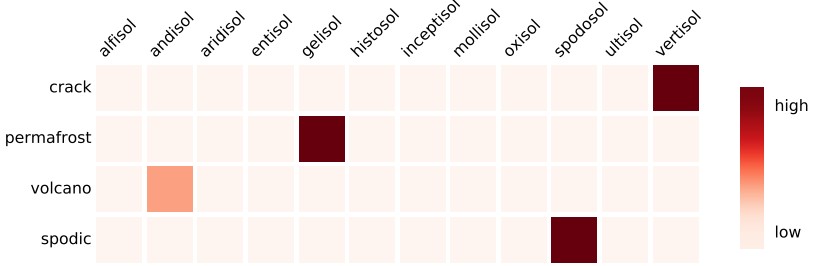

**Figure 3.** Co-occurrence probability matrix of soil orders (USDA) and selected words.

This information can also be used to guide the process of generating a domain-specific model. In our case, in an early stage of this study, the terms "permafrost" and "gelisol" presented a very low co-occurrence probability, a clear sign of the limited topic coverage of the articles at that point.

## 6.2 Intrinsic evaluation

The results of the intrinsic evaluation indicate that our domain-specific embeddings (GeoVec) performed better than the general domain embeddings in all tasks (Table 2), increasing the overall performance by 107.9%. This is an expected outcome considering the specificity of the tasks. For the analogies, we decided to present the top-1, 3, 5 and 10 accuracy scores because, even if the most desirable result is to have the expected word as the first output from the model, in many cases the first few words are closely related or they are synonyms. For instance, for the analogy "fan is to fluvial as estuary is to ____? (coastal)", the first four alternatives are "tidal", "river", "estuarine", "coastal", which are all related to a estuary.

In the relatedness task, the 3 human annotators had a high inter-annotator agreement (multi-kappa=98.66%; as per Davies and Fleiss (1982)), which was expected since the relations are not complex for some with a background in geosciences. As we keep working on this topic, we plan to extend the test suite with more subtle relations.

It was possible to observe an increase on the overall performance of the embeddings (calculated as the mean of the analogy (top-5), relatedness and categorisation tasks) as we added more articles, almost stabilising around 300 million tokens, especially for the analogy task (Fig. 4). For domain-specific embeddings, this limit most likely varies depending on the task and domain.

**Table 2.** Evaluation scores for each task for our domain-specific (GeoVec) and general domain embeddings (Stanford). For the analogy task, top-1, 3, 5 and 10 represents the accuracy if the expected word was within the first 1, 3, 5 or 10 words returned by the model. For the relatedness task, the score represents the absolute value of the Pearson correlation (mean of the 3 human subjects). For the categorisation task, the score represents the mean value of 50 v-measure scores. The possible range of all scores is 0 to 1, where higher is better.

|                   | GeoVec | Stanford |
|-------------------|--------|----------|
| Analogy (top-1)   | 0.39   | 0.22     |
| Analogy (top-3)   | 0.78   | 0.37     |
| Analogy (top-5)   | 0.90   | 0.41     |
| Analogy (top-10)  | 0.92   | 0.49     |
| Relatedness       | 0.61   | 0.23     |
| Categorisation    | 0.75   | 0.38     |
| Overall           | 0.73   | 0.35     |

For instance, Pedersen et al. (2007), measuring semantic similarity and relatedness in the biomedical domain, found a limit of around 66 million tokens.

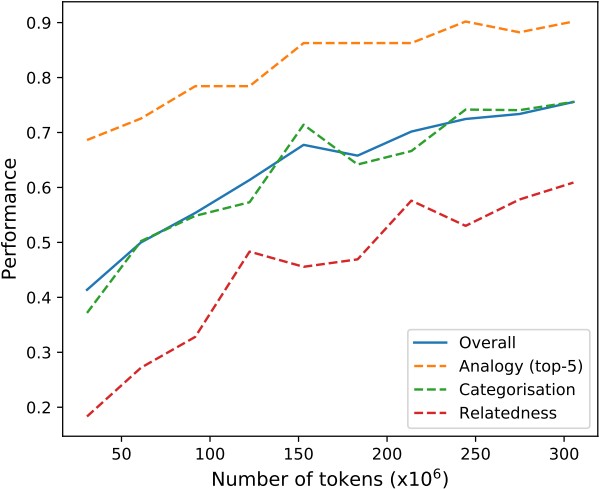

**Figure 4.** Overall performance of the embeddings versus number of tokens used to construct the co-occurrence matrix. The improvement limit is around 300 million tokens. For future comparisons, this limit corresponds to approximately: 280,000 articles, 22.5 million sentences and 700,000 unique tokens.

The improvement over the general domain embeddings has also been reported in other studies. Wang et al. (2018) concluded that word embeddings trained on biomedical corpora can capture the semantics of medical terms better than the embeddings of a general domain GloVe model. Also in a biomedical application, Jiang et al. (2015) and Pakhomov et al. (2016) reported similar

conclusions. In the following sections, we explore the characteristics of the obtained embeddings, showing some graphical examples of selected evaluation tasks.

## 6.3 Analogy

A different way of evaluating analogies is to plot the different pairs of words in a 2-dimensional PCA projection. Fig. 5 shows
different pairs of words which can be seen as group analogies. From the plot, any pair of related words can be expressed as an analogy. For example, from the left panel, it is possible to generate the analogy "claystone is to clay as sandstone is to ____? (sand)" and the first model output is indeed "sand".

As we showed in Fig. 2, the embeddings encode different relationships with different degrees of sophistication. In the left panel of Fig. 5 it is possible to observe simple analogies, mostly syntactic since "claystone" contains the word "clay". The
right panel presents a more advanced relationship where rock names are assigned to their corresponding rock type.

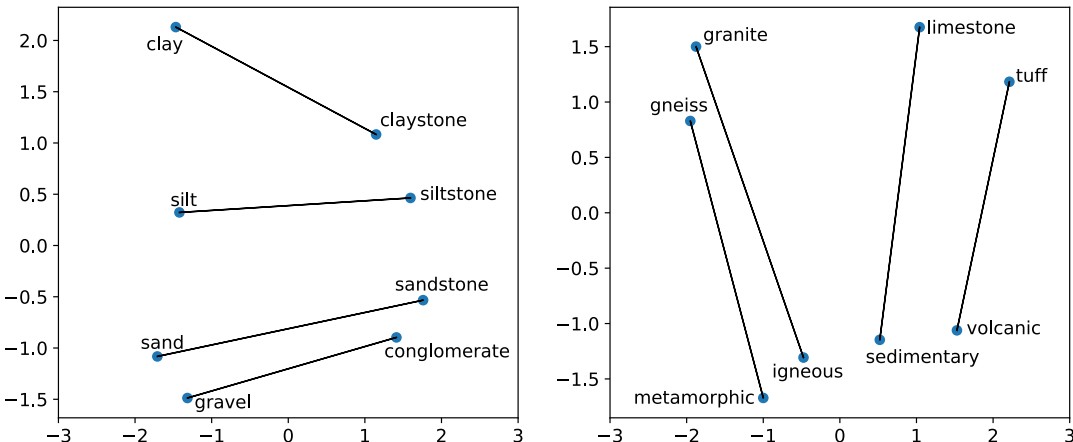

**Figure 5.** Two-dimensional PCA projection of selected words. Simple syntactic relationship between particle fraction sizes and rocks (left panel) and advanced semantic relationship between rocks and rock types (right panel).

## 6.4 Categorisation

Similar to the analogies, the categorisation task can also present different degrees of complexity of the representations. In the left panel of Fig. 6, k-means clustering can distinguish the two expected clusters of concepts, WRB (FAO, 1988) and Soil Taxonomy (USDA, 2010) soil classification names. Andisols and Andosols are correctly assigned to their corresponding groups
but apart from the rest, probably due to their unique characteristics. Vertisols are correctly placed in between the two groups since both have a soil type with that name. A second level of aggregation can be observed in the right panel. The k-means clustering correctly assigned the same soil groups from the left panel into a general "soil types" group, different from "rocks".

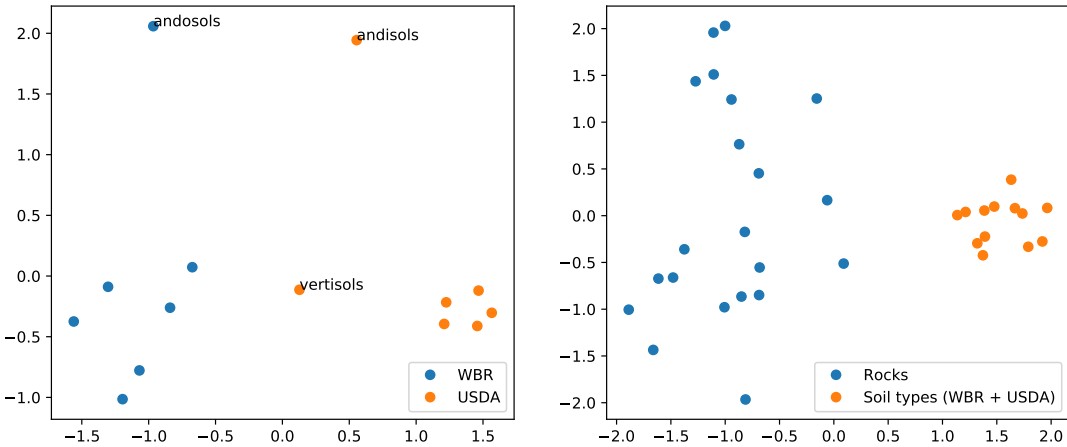

**Figure 6.** Two-dimensional PCA projection of selected categorisations. Clusters representing soil types from different soil classification systems (left panel) and a different aggregation level where the same soil types are grouped as a single cluster when compared with rocks (right panel).

### 6.5  Other embedding properties

Interpolation of embeddings is an interesting exercise that allows to further explore if the corpus is well represented by the vector space. Interpolation has been used to generate gradient between faces (Yeh et al., 2016; Upchurch et al., 2017), assist drawing (Baxter and ichi Anjyo, 2006) and transform speech (Hsu et al., 2017). Interpolation between text embeddings is less common. Bowman et al. (2015) analysed the latent vector space of sentences and found that their model was able to generate coherent and diverse sentences when sampling between two embeddings. Duong et al. (2016) interpolated between embedding from two vector spaces trained on different languages corpora to create a single cross-lingual vector space. The vector space from our model also presents similar characteristics.

In order to generate the interpolated embeddings, we obtained linear combinations of two words embeddings by using the formula

$$v_{int} = \alpha * v_a + (1 - \alpha) * v_b \tag{4}$$

where $v_{int}$ is the interpolated embedding, $v_a$ and $v_b$ are the embeddings of the two selected words. By varying the value of $\alpha$ in the range $[0, 1]$, we generated a gradient of embeddings. For each intermediate embedding obtained by interpolation, we calculated the cosine similarity (Eq. 3) against all the words in the corpus and selected the closest one.

The results showed coherent concepts along the gradients (Fig. 7). The interpolation between "clay" and "boulder", with fine and coarse size, respectively, yields a gradient of sizes, with "clay"<"silt"<"sand"<"gravel"<"cobble"<"boulder". Another interpolation example, along another type of relationship, is shown in the right panel of Fig. 7. The interpolation between the rocks "slate" and "migmatite" yields a gradient of rocks with different grades of metamorphism, with "slate"<"phyllite"<"schist"<"gneiss"<

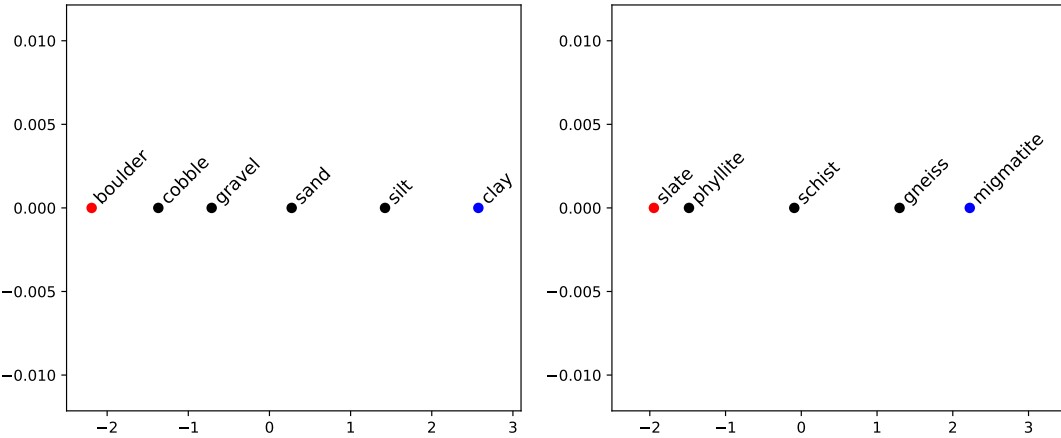

**Figure 7.** Interpolated embedding in a two-dimensional PCA projection showing a size gradient (left panel) with "clay"<"silt"<"sand"<"gravel"<"cobble"<"boulder"; and gradient of metamorphism grade (right panel) with "slate"<"phyllite"<"schist"<"gneiss"<"migmatite". Red and blue dots represent selected words ("clay" and "boulder", and "slate" and "migmatite") and black dots represent the closest word (cosine similarity) to the interpolated embeddings.

### 6.6 Illustrative example

As a final, external evaluation of the embedding, we estimated average embeddings for each Great Group (Soil Taxonomy) of soils from 10,000 soil profiles descriptions. The convex-hulls at soil order level (Fig. 8) show the same pattern reported by Hughes et al. (2017). Thanks to the unique characteristic of Histosols and the high diversity of this taxonomic group, they are easily differentiated in the 2-dimensional projection, showing the highest variability. The rest of the soil orders are heavily overlapped since their differences are hard to simplify into a 2-dimensional space. That overlap does not imply that the orders are not separable in a higher dimensional space. Here we are plotting the first 2 principal components (PCs), which only account for 28.8% of the total variance. This is probably the same reason for the overlap in the study by Hughes et al. (2017) since they account for a 95% of the total variance only after 36 PCs (i.e. their plot, also using the first 2 PCs, probably explain a low proportion of the total variance, similar to our example).

This example shows how, by using descriptions encoded as word embeddings, we were able to use the same methods used by Hughes et al. (2017). In this case, if no soil variables (laboratory data) were available, word embeddings could be used instead. Ideally, we would expect to use word embeddings to complement numerical data to utilise valuable information included in the descriptive data. This is also possible with other approaches. Hughes et al. (2017) manually generated binary embeddings for the presence of ice and water. Another alternative to create embeddings is fuzzy logic. For example, McBratney and Odeh (1997) fuzzify categorical information from soil profiles such as depth, generating an encoding that represents the probability to belong to different depth classes (e.g a "fairly deep" soil could lay between the "shallow" and "deep" classes, with a membership of 0.5 to each class). The advantage of using word embeddings is that they are high dimensional vectors that encode much more information applicable to many tasks, which would be difficult to replicate by manual encoding.

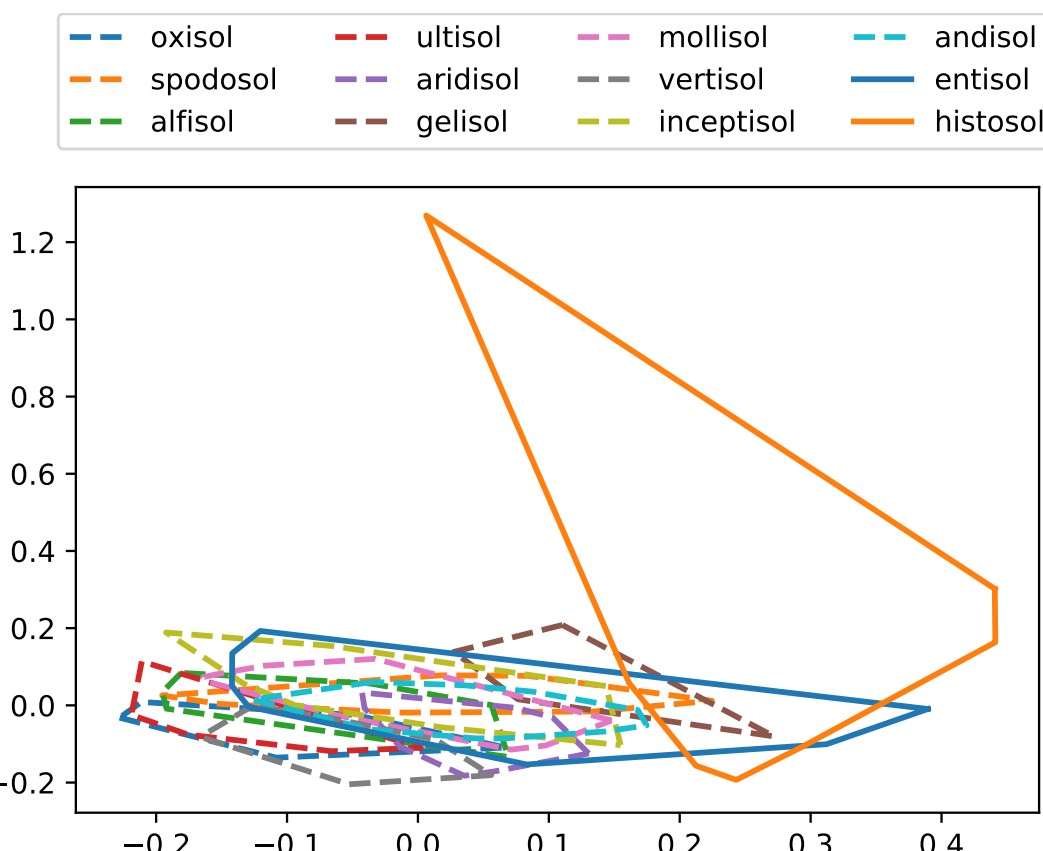

**Figure 8.** Convex-hulls of great group embeddings at the order level (Soil Taxonomy). Great group embeddings were obtained after averaging the embeddings of all the words in the descriptions of the profiles belonging to each great group. The convex-hulls were estimated from the 2 first principal components of the great group embeddings.

### 6.7 What do these embeddings actually represent?

It is worth discussing if word embeddings tell us anything about nature or if they really just tell us about the humanly constructed way that science is done and reported. A language model extracts information from the corpora to generate a representation in a high dimensional space. This continuous vector space shows interesting features that relate words to each other, which were tested in multiple tasks designed to evaluate the syntactic regularities encoded in the embeddings. Considering the position that science is a model of nature (Gilbert, 1991) and assuming that the way we do and report science is a good representation of it, if the language model is a good representation of the corpora of publications, perhaps the derived syllogism — the language model is a good representation of nature — can be considered as true. Of course, the representation of a representation carries many impressions, but it is worth exploring its validity.

As shown by the linear combinations of embeddings (Fig. 7), some aspects related to "size" are captured by the embeddings and, even if size categories are a human construct, they describe a measurable natural property. A more complex case is the illustrative example, where the embeddings capture some aspect of nature which are also captured by the numerical representation of its properties (in this case soil properties such as clay content, pH, etc). Given the results of the intrinsic evaluation of this work and others referenced throughout this article, it is probably impossible to generate the "perfect embeddings". Even if we were able to process all the written information available, and ignoring the limitations of any language model, the embeddings would be still limited by our capacity to understand non-linear relationships (Doherty and Balzer, 1988) and, in consequence, to understand nature.

Whether word embedding can give new insights about geosciences is still to be tested. Studies in other fields have shown some potentially new information. For instance, Kartchner et al. (2017) generated embeddings from medical diagnosis data and, after performing a clustering, they found clear links between some diagnoses related to advanced chronic kidney disease. Some of the relations are already known and accepted by the medical community while others are new and are just starting to be studied and reported.

### 6.8  Future work

In the future, we expect to evaluate the effect of using our embeddings in more downstream applications (extrinsic evaluation). It is expected that domain-specific embedding will necessarily improve the results of downstream tasks but this is not always the case. Schnabel et al. (2015) suggested that extrinsic evaluation should not be used as a proxy for a general notion of embedding quality, since different tasks favour different embeddings, but they are useful in characterising the relative strengths of different models. We also expect to expand the test suite with more diverse and complex tests, opening the process to the scientific community. Another interesting opportunity is the inclusion of word embeddings in numerical classification systems (Bidwell and Hole, 1964; Crommelin and De Gruijter, 1973; Sneath et al., 1973; Webster et al., 1977; Hughes et al., 2014) which try to remove subjectivity by classifying an entity (soil, rock, etc.) based on numerical attributes that describe its composition.

### 7  Conclusions

In this work we introduced the use of domain-specific word embeddings for geosciences (GeoVec), and specifically soil science, as a way to a) reduce inconsistencies of descriptive data, and b) open the alternative to include such data into numerical data analysis. Comparing the result with general domain embeddings, trained on corpus such as Wikipedia, the domain-specific embedding performed better in common natural language processing tasks such as analogies, terms relatedness and categorisation, improving the overall accuracy by 107.9%.

We also presented a test suite, specifically designed for geosciences, to evaluate embedding intrinsic performance. This evaluation is necessary to test if syntactic or semantic relationships between words are captured by the embeddings. The test suite comprises tests for three tasks usually described in the literature (analogy, relatedness and categorisation) with different levels of complexity. Since creating a set of gold standard tests is not a trivial task, we consider this test suite a first approach.

In the future, we expect to expand the test suite with more diverse and complex tests and to open the process to the scientific community to cover different subfields of geosciences.

We demonstrated that the high-dimensional space generated by the language model encodes different types of relationships, through examples of soil-related concepts. These relationships can be used in novel downstream applications usually reserved for numerical data. One of these potential applications is the inclusion of embeddings in numerical classification. We presented an example were we successfully emulated part of a taxonomic analysis of soil profiles which was originally applied to soil numerical data. By encoding soil descriptions as word embeddings we were able to use the same methods used in the original application and obtain similar results. Ideally, we would expect to use word embeddings when no numerical data is available or to complement numerical data to include valuable information included in the descriptive data.

*Code availability.* The embeddings, test suite and helper functions will be available at https://github.com/spadarian/GeoVec

*Competing interests.* The authors declare that they have no conflict of interest.

*Acknowledgements.* This research was supported by Sydney Informatics Hub, funded by the University of Sydney.

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
