# Peer review of "Word embeddings for application in geosciences: development, evaluation and examples of soil-related concepts"

_SOIL, 2018_

## Referee Comment (RC1) · Diana Maynard (Referee) · 9 Feb 2019

In this paper, the authors train some standard word embeddings specifically on a geoscience corpus and show that, unsurprisingly, these are better than some pre-existing embeddings trained on a general corpus. This is well-known, so there are no particularly interesting findings specifically in that result. Making the embeddings available to others could be beneficial, but for this a larger set would ideally be required, and certainly more proof of their usefulness would be needed.

While the application of word embeddings-based language analysis techniques is relatively new in the field of geo-sciences, the authors do not provide sufficient scientific

contribution or motivation in this paper. I have two major issues with the work presented: 1. It's not clear what they want to use word embeddings specifically for. They experiment with training some existing techniques on a geoscience corpus, but there is no actual motivation for doing so. Word embeddings are only useful if they are applied to a specific task, and if it can be shown that they help to solve the task in a better way than existing techniques. But the authors give no real-world task and just evaluate the quality of the embeddings on standard fun tasks such as analogies that have no actual purpose. Creating a good set of embeddings is one thing, but half the task lies in finding the best way to transform the topic vectors. 2. The authors do not do anything that involves scientific novelty - they simply take some existing word embeddings models and train them on a new corpus, which requires no novel critical thinking. This is useful therefore only as a means to an end, but is not worthy of publication in itself. The work could be interesting if it were taken a step further, but currently it is insufficient.

Some more specific points follow.

The abstract does not make it clear what the motivation for the work is, beyond the fact that word embeddings have not been trained on such a corpus, but this is insufficient justification.

The introduction is vague, e.g. "different machine learning methods have been used for geoscience" - what does this tell us? Nothing. We need to know at least what task they have been used for, why they have been used, and how well they work, not to mention why it is relevant to the work presented. Similarly, much of it is too imprecise, e.g. "subjectivity and ambiguity introduced by language can be removed by text processing and probabilistic analysis" - this needs clarification. Subjectivity can almost never be "removed" by NLP techniques, and as for ambiguity, this depends a lot on the kind of ambiguity and the task - and typically only those things which are ambiguous to computers but not humans can be dealt with successfully.

Again, the introduction should explain the motivation behind the work, why word embeddings are useful, and give some idea of the kinds of tasks they are going to be used for here. References to related work are lacking - the authors need to do proper research into the state of the art here - for example, properly investigating the advantages/disadvantages of training word embeddings on a general vs specific corpus.

The section on word embeddings is neither a clear general explanation for those who have no idea what they are (as one might expect in the geoscience field), nor does it provide a technical explanation for those familiar with the topic. The authors introduce the idea of analogies being produced with word embeddings, but do not explain why this is even interesting. Figures 1 and 2 are not clear to those who don't know already about word embeddings, and obsolete for those who do. In general, this section is very inadequate.

Section 3 is lacking in technical detail. How were the terms listed in Table 1 decided? Why were these particular pre-processing decisions taken? See for example (Denny and Spirling, 2018) on the importance of such decisions on the results obtained, and the effect that even small changes to these decisions can have on the end results. Denny, Matthew J., and Arthur Spirling. "Text preprocessing for unsupervised learning: why it matters, when it misleads, and what to do about it." Political Analysis 26.2 (2018): 168-189. For example, why do you use stemming and not morphological analysis? Surely you do not want to conflate tokens with different POS tags here? In other words, you want to perform inflectional but not derivational morphological analysis - this is more commonly used for pre-processing word embedding training than just stemming (the easy option). Either way, these decisions need to be properly justified.

Evaluation: You need to provide proper information here. For the relatedness task, who did the scoring? Was Inter-Annotator Agreement measured (and if not why not?)? What was the result of IAA? I would not expect high agreement here because this is a hard task for humans to perform, so this is really critical in order to have a valid set of gold standard data. How many tokens is your dataset also?

Section 5.5 - I suggest explaining what you mean by interpolation of embeddings.

The Conclusion section is very brief and, unsurprisingly given the rest of the paper, gives no real interesting conclusions. The final sentence is very unsatisfactory: what do you even mean by saying that embeddings give the scientific community an interesting way of "exploring how a scientific community creates its own language...."? You certainly haven't studied this in this work, and have no insights to offer us on it.

In summary, the development of a specific set of embeddings for geosciences could be useful, but this is all rather insufficient for publication here, and the relevance to geosciences, and specifically soil, is minimal. I suggest waiting until you have at least attempted to resolve some specific task using the embeddings, which relevant specifically to the topic of soil, before attempting to publish.

---

## Short Comment (SC1) · 22 Feb 2019

This study discusses the potential of word embeddings to include descriptive information in geosciences. Although I found this an interesting approach, that could be particularly valuable for soil science, I struggled somewhat to see how this manuscript exactly demonstrates its potential. The conclusion that a domain-specific embeddings outperforms a general domain embeddings has been reported before. The development of the domain-specific GeoVec embedding is an interesting starting point/tool for new research and the linguistic relations captured by the model (as shown in figure 5-7) make sense. However, in my opinion, the study could have a much bigger impact if it could describe and demonstrate how we can valorize the large amounts of descriptive information into quantitative soil science. The authors make some valuable suggestions in the 'Future work' section that could exactly do this.

Other comments: Section 3.2. Is the approach robust? It could be interesting to see how these specific pre-processing steps influence the model performance and relations. Figure 3: a relative scale is used. Can this be quantified? Section 5.2: why is there such a big difference in performance stabilization between geosciences and biomedical sciences? What does the color code represent in figure 7¿

––––––––––––––––––––––––––––––

---

## Short Comment (SC2) · 23 Feb 2019

First of all it is worth establishing the basis of the analysis. As I understand it - it seems to be based on the multidimensional scaling (principal coordinates) of a co-occurrence probability matrix - where the co-occurrences are related to a short list of pairs of words =(terms?) separated by a given word distance The words being meaningful in the soil science community. Is this a reasonable summary?

I guess one can ask what can we gain from this kind of analysis? It is very dependent on the words chosen. However it does show which words, and therefore concepts?, are used together. Does it reflect a pattern of usage? a similarity or difference of

concepts? or something more? From this can we quantify papers in terms of their content? Textual analysis is used to quantify free-form responses to survey questions. Scientific papers are of course less free form. I guess in the end my main question is - does this kind of analysis tell us anything about science or nature or does it really just tell us about the humanly constructed way that science is done and reported?

Looking at Figure 6. The diagram on the left looking at the names of soil 'orders' or 'reference groups' in a couple of systems shows that the two systems do not overlap - that the word 'vertisol' used in both systems split the difference - but of course the meaning/definition in the two systems might not be the same. It also suggests that volcanic soils are different in the two systems but are different from other soils. What do we learn - is there no overlap between the two systems? - if so then this is a complete disaster for soil classification. One interpretation is that it shows that there are two user communities and they do not cross-reference each other.Unhelpful for soil science and soil sustainability. The recent quantitative work in Geoderma by Hughes et al based on soil properties shows some degree of overlap between systems . It would be useful to label all the points on this diagram. The diagram on the right shows no overlap between soil classes and rock classes. It might suggest also that soils are more similar to each other than rocks, or that the way the words are used are more heterogeneous.

I could not quite understand Figure 7 - it shows meaningful continua of terms and in the correct order - is it a construction? or is it based on an analysis of papers? This reminds me of course of another approach - if one of the aims of the work here is to attempt to quantify meaning via words - then the fuzzy or continuous class approach is a good alternative, and perhaps should be compared. There are papers by the late Inakwu Odeh on this, and part a chapter in the recently-published Pedometrics book.

---

## Referee Comment (RC2) · Anonymous Referee #2 · 28 Feb 2019

This study discusses the potential of word embeddings to include descriptive information in geosciences. Although I found this an interesting approach, that could be particularly valuable for soil science, I struggled to see how this manuscript exactly demonstrates its potential. The conclusion that a domain-specific embeddings outperforms a general domain embeddings has been reported before. The development of the domain-specific GeoVec embedding is an interesting starting point/tool for new research and the linguistic relations captured by the model (as shown in figure 5-7) make sense. However, in my opinion, the study could have a much bigger impact if it could describe and demonstrate how we can valorize the large amounts of descriptive information into quantitative soil science. I think this is a critical issue that needs to be

addressed. The authors make some valuable suggestions in the 'Future work' section that could exactly do this.

Other comments: Section 3.2. Is the approach robust? It could be interesting to see how these specific pre-processing steps influence the model performance and relations. Figure 3: a relative scale is used. Can this be quantified? Section 5.2: why is there such a big difference in performance stabilisation between geosciences and biomedical sciences? What does the color code represent in figure 7?

---

## Author Comment (AC1) · 2 Mar 2019

Thanks for all the comments. We are sure they will greatly improve the manuscript.

Here we discuss some points that we would like to add to the manuscript, following the reviewers' comments.

[Figure]

**1 Level of detail**

We are happy to address the points raise by Diana Maynard about giving more details about the methodology and the rationale behind some of the decisions taken. This will also help to delimit the scope of the manuscript. We are trying to introduce the concept of word embeddings to the geosciences audience and document the process of generating the embeddings, including their evaluation in tasks such as analogies, relatedness, and categorisation, which seems to be a widely used method to assess the linear substructures generated by the model.

We think some of the points raised by the reviewers are beyond the scope of the manuscript. For instance, how different pre-processing steps influence the model performance and relations. We tried some of the "pipelines" commonly used in the NLP literature instead of comparing all the possible combinations (which, according to a paper suggested by Diana, could be hundreds). The representations that we observed made sense for us. We are not linguists or ontology experts, and we could be biased, but probably as much as any external expert in the field of geosciences.

**2 Illustrative example**

We realise that an example will greatly improve the manuscript, as pointed out by Diana Maynard and Kristof Van Oost. We would like to comment on the inclusion of an example.

- In other fields, word embeddings (specifically generated for the task or, in many cases, the same general embeddings) are used in a plethora of applications. They have been proven useful in diverse areas so we think there is a general consensus about their applicability.

- According to part of the NLP literature, extrinsic evaluation of embeddings (using embeddings in a downstream task) is not a good indicator of their quality. Of course, we agree that they need to be useful for something, but the range of applications is wide and the complexity of creating gold standard downstream tasks datasets is high.

**3   Proposed example**

(Just an example and not the final text) We downloaded around 10,000 soil profile descriptions from the USDA-NRCs Web Site for Official Soil Series Descriptions and Series Classification. To each profile, we applied the same pre-processing performed when generating the model (tokenisation, lemmatisation, etc.). After obtaining the embeddings for each token, we calculated the mean values per profile, which can be considered as the encoding at profile level. Each profile and its corresponding 300-dimensional encoding were aggregated at Great Group (GG) level (Soil Taxonomy) and a mean embedding value was estimated. After projecting the GG embeddings in a PCA space, we computed the convex hull per soil order (see attached figure). The resulting figure shows the extent of each order and it is comparable with the results shown by Hughes et al. (2017) who performed a similar exercise but with soil properties (OC, sand, clay, etc) in the context of a numerical soil classification system. Similar to the results reported by Hughes et al. (2017) Histosols are easily differentiable and they show relatively high variability.

Hughes, P., McBratney, A.B., Huang, J., Minasny, B., Micheli, E. and Hempel, J., 2017. Comparisons between USDA Soil Taxonomy and the Australian Soil Classification System I: Data harmonization, calculation of taxonomic distance and inter-taxa variation. Geoderma, 307, pp.198-209.

**4 Specific comments to Kristof Van Oost**

**Figure 3: a relative scale is used. Can this be quantified?**

Yes. We generated the relative scale from numerical data. We think the relative scale illustrates well the word co-occurrence in our corpus.

**Section 5.2: Why is there such a big difference in performance stabilization between geosciences and biomedical sciences?**

We don't think both studies are directly comparable. As we mention in the manuscript, it's probably dependent on the tests, domain, corpora used, etc.

**What does the colour code represent in figure 7?**

The two extreme terms (red and blue) are the 2 words used in the interpolation and the black terms are the interpolated terms. We will explain that plot better. The interpolation corresponded to a linear combination of the 2 extreme encodings

$$v_{int} = \alpha * v_a + (1 - \alpha) * v_b \tag{1}$$

where $v_{int}$ is the interpolated embedding, $v_a$ and $v_b$ are the embeddings of the selected words. We varied the value of ðİŽij in the range [0, 1] to obtain the figures.

[Figure]

**5 Specific comments to Alex McBratney**

**First of all it is worth establishing the basis of the analysis. As I understand it - it seems to be based on the multidimensional scaling (principal coordinates) of a co-occurrence probability matrix - where the co-occurrences are related to a short list of pairs of words=(terms?) separated by a given word distance The words being meaningful in the soil science community. Is this a reasonable summary?**

The analysis is based on the co-occurrence matrix but the model is more complex than a multidimensional scaling. We will expand the model description to clarify this.

**Does this kind of analysis tell us anything about science or nature or does it really just tell us about the humanly constructed way that science is done and reported?**

This is a very good question. These kinds of models extract information from the corpus to generate a representation in a high dimensional space. From a linguistic point of view, this "model" shows interesting features of the text based on words co-occurrence. Assuming that this model is a good representation of the corpus and that the way we report and do science is a good representation of nature, maybe we can assume that the derived syllogism (that the language model is a good representation of nature) is true.

The newly added example shows that the embeddings capture some aspect of nature which are also captured by the numerical representation of their properties (clay, SOC, etc). Also, the interpolation shows some aspects related to size, which, even if the categories are a human construct, describe a measurable natural property. Of course, the representation of a representation carries many impressions, but it is worth exploring it.

**I could not quite understand Figure 7 - it shows meaningful continua of**

**terms and in the correct order - is it a construction? or is it based on an analysis of papers? This reminds me of course of another approach - if one of the aims of the work here is to attempt to quantify meaning via words - then the fuzzy or continuous class approach is a good alternative, and perhaps should be compared.**

We will expand the interpolation section to clarify that. See comments to Kristof Van Oost.

Continuous and fuzzy classes are another example of encoding and we will mention them in Section 2. The main difference is that they are manually generated for a target class, but this approach generates the embeddings "automatically" based on the corpus. Of course, they are, at best, as good as the corpus and probably some relationships are missing, but for sure they include relationships that are hard to encode manually.

**6 Specific comments to Diana Maynard**

**1. It's not clear what they want to use word embeddings specifically for. They experiment with training some existing techniques on a geoscience corpus, but there is no actual motivation for doing so. Word embeddings are only useful if they are applied to a specific task, and if it can be shown that they help to solve the task in a better way than existing techniques.**

We are adding an example to solve this.

**"... [the authors] just evaluate the quality of the embeddings on standard fun tasks such as analogies that have no actual purpose."**

We think it depends on how far we want to take the definition of purpose. By reading the literature, it seems that many NLP researchers focus on developing "fun" analogies, relatedness, and categorisation tasks in order to evaluate word embeddings. Those tasks are designed to test the syntactic regularities encoded in the embeddings, describing how well the generated multi-dimensional space represents the corpus. We would say that that is a well defined purpose.

We designed a set of tests to perform the aforementioned tasks, and we agree that that will not revolutionise the NLP word, but it is something that has to be done in order to create good embeddings.

**6.1 Specific comments**

**- 'The introduction is vague, e.g. "different machine learning methods have been used for geoscience" - what does this tell us? Nothing. We need to know at least what task they have been used for, why they have been used, and how well they work, not to mention why it is relevant to the work presented.'**

We think it is irrelevant what, why and how machine learning methods have been used, except for the fact that they prioritise numerical data over qualitative descriptions. Nevertheless, we will expand this section giving some examples.

**- "References to related work are lacking - the authors need to do proper research into the state of the art here - for example, properly investigating the advantages/disadvantages of training word embeddings on a general vs specific corpus."**

In the introduction, we give two examples of studies where models trained on a specific corpus which conclude that they can capture the semantics of domain-specific terms better than those trained on general corpora. We also mention another reference in the results and discussion section. A quick search yielded 3 more examples that conclude the same, in different fields (which we will add). This seems to be in line with the reviewer's comment on the first paragraph of her review ("...unsurprisingly, these are better than some pre-existing embeddings trained on a general corpus"). We think that 6 references are enough.

**- "The section on word embeddings is neither a clear general explanation for those who have no idea what they are (as one might expect in the geo-science field) nor does it provide a technical explanation for those familiar with the topic. The authors introduce the idea of analogies being produced with word embeddings, but do not explain why this is even interesting. Figures 1 and 2 are not clear to those who don't know already about word embeddings, and obsolete for those who do. In general, this section is very inadequate."**

We will modify this section accordingly to reach the target audience.

**- "Section 3 is lacking in technical detail. How were the terms listed in Table 1 decided? Why were these particular pre-processing decisions taken? See for example (Dennyand Spirling, 2018) on the importance of such decisions on the results obtained, and the effect that even small changes**

**to these decisions can have on the end results. Denny, Matthew J., and Arthur Spirling. "Text preprocessing for unsupervised learning:why it matters, when it misleads, and what to do about it." Political Analysis 26.2 (2018):168-189. For example, why do you use stemming and not morphological analysis? Surely you do not want to conflate tokens with different POS tags here? In other words,you want to perform inflectional but not derivational morphological analysis - this is more commonly used for preprocessing word embedding training than just stemming(the easy option). Either way, these decisions need to be properly justified."**

We will add more technical details and more insights about the different steps. We utilised method widely used in NLP literature and we will add some references to help the reader.

Just as a comment, we didn't observe a difference when using morphological analysis, and definitely didn't change the interpretation of the embeddings. The structure of most of the descriptive data that we mention (pedon and auger descriptions) is simple and based on the occurrence of something specific. For instance, If we observe the occurrence of weathering, the description probably includes the word "weathering", "weathered", and that both have a different POS tag (part-of-speech tag, e.g.: verb, adjective) does not change the results. And of course, we preferred the "easier" (simpler) solution.

**- "Evaluation: You need to provide proper information here. For the relatedness task, who did the scoring? Was Inter-Annotator Agreement measured (and if not why not?)? What was the result of IAA? I would not expect high agreement here because this is a hard task for humans to perform, so this is really critical in order to have a valid set of gold standard data. How many tokens is your dataset also?"**

We will add more information and thanks for the suggestions. We did not measure IAA, but we will add that information. We actually expect a relatively high agreement since the task was performed by people with a background in geosciences and the relations are not extremely complex (for people with some training). We are sure that giving some examples and providing more information, as the reviewer suggests, will clarify things.

About the number of tokens, we provide all the information in Fig. 4. Around 300 million tokens, and 700,000 unique tokens. Also under Fig. 4 (section 5.2, intrinsic evaluation), we provide more examples of studies using domain-specific embeddings.

**- "The Conclusion section is very brief and, unsurprisingly given the rest of the paper, gives no real interesting conclusions. The final sentence is very unsatisfactory: what do you even mean by saying that embeddings give the scientific community an interesting way of "exploring how a scientific community creates its own language...."? You certainly haven't studied this in this work, and have no insights to offer us on it."**

You are right about the final sentence and we are happy to remove it.

[Figure]

[Figure]

**Fig. 1.**